# Voltage detected single spin dynamics in diamond at ambient conditions

Sergei Trofimov [1], Klaus Lips[1,2,3] & Boris Naydenov [1,2] ✉

Defect centres in crystals like diamond or silicon find a wide application in quantum technology, where the detection and control of their quantum states is crucial for their implementation as quantum sensors and qubits. The quantum information is usually encoded in the spin state of these defect centres, but they also often possess a charge which is typically not utilized. We report here the detection of elementary charges bound to single nitrogen-vacancy (NV) centres several nanometres below the diamond surface using Kelvin Probe Force Microscopy (KPFM) under laser illumination. Moreover, the measured signal depends on the NV's electron spin state, thus allowing to perform a non-optical single spin readout, a technique we refer to as "Surface Voltage Detected Magnetic Resonance" (SVDMR). Our method opens a way of coherent spin dynamics detection for quantum sensing applications and could be potentially applied to other solid state systems. We believe that this voltage-based readout would help to simplify the design of devices for quantum technology.

Defect centres in semiconductors carrying electron spins, such as donors in silicon[1], $V_2$ centres in silicon carbide (SiC)[2] and nitrogen-vacancy (NV) centres in diamond[3] are promising physical systems for the emerging quantum technology. For luminescent defects optical initialisation and readout of the spin state can be efficiently performed[4], but for device applications electrical readout is more favourable[5].

The conventional electrical readout of spins is based on a spin-dependent current (often a photocurrent)[6]. This method usually requires either low temperatures and high magnetic fields (for donors in silicon[7]) or applying high electric fields (tens of volts across few microns) and high laser intensities for obtaining the necessary photocurrents (for $V_2$ centres in SiC[8] and NVs[9]) to enhance the measured signal. Nevertheless, typically obtained spin-induced changes in the current, even for defect ensembles, are on the order of nA or lower. Therefore, the current is often detected using a high gain transimpedance amplifier, which poses limitations on the detection of fast-changing signals[10]. To overcome this in continuous wave (CW) detection, low-frequency envelope modulation and lock-in techniques are often utilised to perform electrical readout of coherent spin dynamics

(see, for example[11]). For improving the spatial resolution down to the sub-nanometre range, further current detection schemes can be applied such as conductive atomic force microscopy (c-AFM)[12] or scanning tunnelling microscopy (STM)[13].

The electrical current needed for the qubit readout is often generated by photoexcited carriers, but the latter can be also detected using the photovoltage effect[14]. For example, sub-bandgap illumination has been reported to change the diamond surface potential measured in surface photovoltage (SPV)[15] or Kevin Probe Force Microscopy (KPFM) experiments[16]. The observed photovoltage (PV) is usually attributed to charge carrier release from and trapping at surface states located in the diamond band gap[17] and various bulk defects, such as NV centres[18] or silicon-vacancy (SiV) centres[19]. Photoexcitation of surface traps typically leads to the emission of free holes in the diamond valence band[16] (see Fig. 1a, i). In the case of the NV centre under green laser excitation, the carriers of both types (electrons and holes) are produced continuously in a charge-cycling process induced by two two-photon transitions[20] as shown in Fig. 1a, ii and iii. It has been shown that the charge cycling dynamics depend on the NV centre's spin in the ground state due to the spin-dependent inter-system

[1]Berlin Joint EPR Laboratory and Department Spins in Energy Conversion and Quantum Information Science (ASPIN), Helmholtz-Zentrum Berlin für Materialien und Energie, Hahn-Meitner-Platz 1, 14109 Berlin, Germany. [2]Department of Physics, Freie Universität Berlin, Arnimallee 14, 14195 Berlin, Germany. [3]Adjunct Professor, Department of Physics & Astronomy, University of Utah, Salt Lake City, USA. ✉e-mail: boris.naydenov@helmholtz-berlin.de

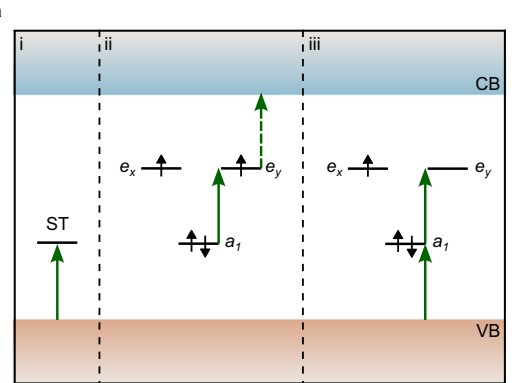
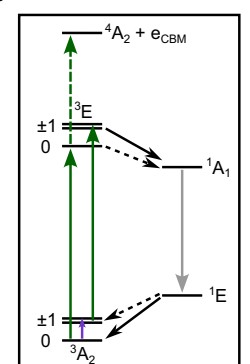
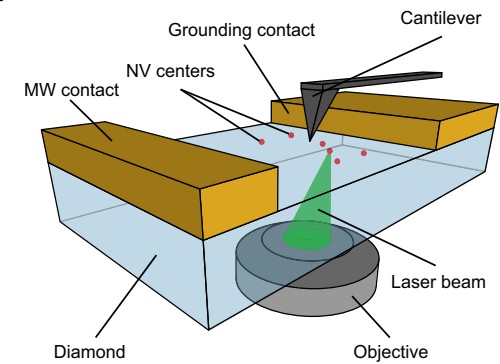

**Fig. 1 | Schematic representation of the conducted experiments. a** Laser-induced transitions (green arrows) in the diamond bandgap. Surface trap (ST) states emit free holes (i), while NV centres undergo a charge cycling process (ii − NV⁻ → NV⁰ and iii − NV⁰ → NV⁻), resulting in the creation of charge carriers of both types in the diamond conduction (CB) and valence (VB) bands. The spin-dependent transition is shown as a dashed arrow. Small black arrows indicate electron spins occupying atomic orbitals in a single-electron picture. **b** Ionization process of the NV⁻ centre dependent on the spin projection (0 and ±1 states). Labels denote the

NV centre energy levels; $e_{CBM}$ represents an electron at the CB minimum. Green arrows show the laser-induced excitation (solid lines) and the spin-dependent ionization (dashed green transition)[45]. Black arrows indicate inter-system crossing transitions with high (solid lines) and low (dashed lines) probabilities. The microwave transition used here for spin manipulation is marked with a violet arrow. A solid grey arrow indicates the infrared transition between singlet states. The energy distances are not to scale. **c** Schematic of the experimental setup, adapted from ref. [43].

crossing[11,21]. Namely, for $m_s = \pm 1$, the higher probability to undergo intersystem crossing (compared to $m_s = 0$) translates to the lower number of photoexcited free carriers, because of the low singlet state ionisation cross-section (see Fig. 1b). These free carriers could then be trapped by other nearby defects, hence converting the spin information to a change of the charge state, which is usually probed via photoluminescence (PL). This method was successfully demonstrated for ensembles of NV (spin-dependent hole source) and SiV (hole trap[22]) centres in diamond[23] as well as for pairs of single NV centres, where one defect served as a hole source under laser illumination, while the other one acted as a hole trap[24]. In those experiments, spin state detection relies on photoluminescence-active defects in the bulk created either during diamond growth or by electron/ion implantation. Diamond surface states, on the other hand, are also able to trap free carriers[25,26], do not affect the bulk crystalline structure and can be altered using different surface treatment procedures[27,28]. Charges, trapped in these states, change the electrical potential of the surface[29] and can be indirectly probed by measuring the surface potential in KPFM or SPV experiments, thus, in principle, allowing to use them for detecting the spin state of defects within the band gap.

In frequency-modulated KPFM experiments[30], a probe oscillates at a frequency $f_0$ in the vicinity of the electrically grounded sample surface. An alternating current (AC) potential with an amplitude $V_{AC}$ at a frequency $f_{AC}$ applied to the probe induces mechanical oscillations at sideband frequencies $f_0 \pm f_{AC}$. The amplitude of these sideband oscillations is proportional to the contact potential difference (CPD) between the probe (cantilever) and the sample and is detected using a lock-in technique. A voltage feedback loop is used to apply a direct current (DC) potential to the probe in order to nullify the sideband oscillations, thus measuring the CPD that reflects the electrical potential of the surface.

Here, we utilise this idea and present a method for electrical readout of electron spins associated with defect centres in solids based on changes of the surface potential using KPFM under laser illumination. As a test system we use NV centres in diamond located a few nanometres below the surface.

## Results

To demonstrate the effect described above, we perform a KPFM-based readout of NV centres to detect their spin state. A schematic of the experiment setup is shown in Fig. 1c. A thin electronic grade diamond

plate (50 µm thickness) with shallow (7 nm below the surface) single NV centres created by nitrogen implantation (energy 5 keV, dose $5 \cdot 10^9$ ions/cm²) was used as a sample. Gold micro-strip lines were deposited on the surface for electrical grounding and for applying microwaves (MW) to control the NV's electron spins. A cantilever was probing the CPD between the diamond surface and the tip in a sideband KPFM mode, while a green (520 nm) laser was focused to a diffraction-limited spot by an objective positioned underneath the sample. The laser spot could be moved laterally relative to the diamond plate and to the cantilever. During imaging experiments, the position of the laser spot that was scanned around the cantilever was correlated with the KPFM signal, leading to CPD images. It is important to note that in contrast to a probe scanning technique, here the cantilever is not moving across the sample surface and changes in the probed signal are induced only by moving the laser spot. To obtain photovoltage (PV) images characterising changes in the sample surface potential induced by the laser illumination, we subtracted from the CPD images the KPFM signal measured beforehand with the laser turned off.

A demonstration of NV centre detection using this technique is shown in Fig. 2b, where the negative PV is encoded in blue, while the positive PV is brown. The separated spots in the image represent single NV centres, which is confirmed by comparison with the simultaneously recorded photoluminescence (PL) image (Fig. 2a). The cantilever tip is also visible in the centre of the PV image as a blue spot with a white halo. Figure 2b demonstrates that the diamond surface shows a positive signal $PV_{surf}(x, y)$ that decreases as the laser approaches the tip. The photovoltage signal $PV(x, y)$ thus consists of the surface photovoltage $PV_{surf}(x, y)$ and signal from the NV centres $PV_{NV}(x, y)$, so that $PV(x, y) = PV_{NV}(x, y) + PV_{surf}(x, y)$. We show the $PV_{NV}$ image in Fig. 2c, where each NV centre in Fig. 2b was fitted with a two-dimensional (2D) Gaussian function to obtain the $PV_{NV}(x, y)$ and $PV_{surf}(x, y)$ values. Further details about the data processing can be found in the Supplementary Information (SI). The NV centres in the middle of the scan were not analysed due to their proximity to the cantilever.

Apart from "bright" NV centres, the presented PV image reveals a defect that does not show luminescence. The absence of luminescence could indicate the presence of an NV centre with a proximal local electron trap, such as nitrogen centres, vacancy complex, or other acceptor states (see, for example,[31]). In this case, the NV centre spends more time in the neutral charge state during illumination. The negative PV signal then would originate not from the NV⁻ → NV⁰ transition itself

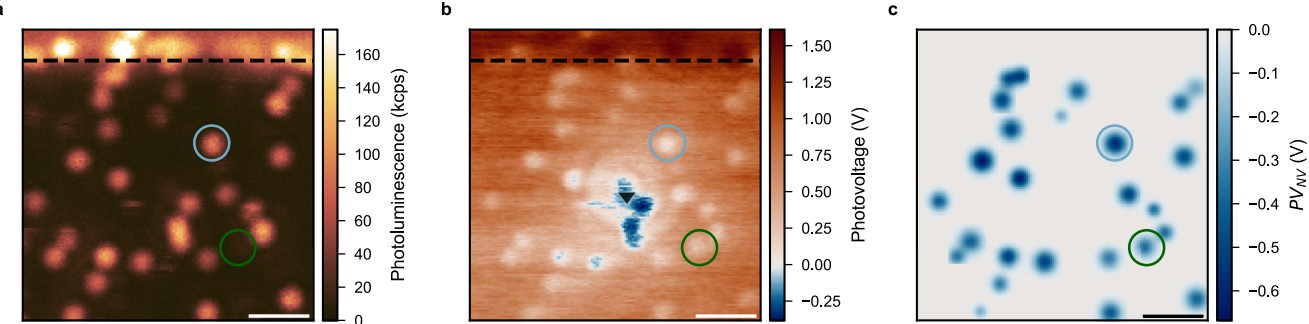

**Fig. 2 | NV centre imaging. a** PL image of single NV centres. **b** PV image obtained simultaneously with (**a**). The colour scale is chosen so that the positive PV is brown, the negative − blue, and the PV equal to zero is white. A defect centre showing no PL but PV contrast is circled in green. An NV centre circled in light blue was used for magnetic resonance and Rabi experiments demonstrated in Fig. 3. The position of the cantilever and the grounding electrode are shown with a black triangle and a dashed line, respectively. The small lateral shift of the PV image compared to the PL image is induced by the high scanning rate and is explained in the SI. **c** $PV_{NV}$ image obtained from fitting (**b**) as explained in the main text. The scale bars are 1 μm. The accumulation parameters are: image size − 150 × 150 pixels, scanning rate − 200 Hz (5 ms per pixel), total scan time − 112.5 s, laser power − 500 μW. The KPFM detection parameters are: AC potential amplitude $V_{AC}$ − 6 V, AC potential frequency $f_{AC}$ − 3 kHz, lock-in time constant − 1 ms, lock-in sensitivity − 1 V, voltage feedback loop gain − 0.5.

**Fig. 3 | NV centre spin state detection. a** SVDMR spectrum from a single NV centre ($PV_{NV}^{off}$ = − 610 mV). **b** Rabi oscillations from a single NV centre obtained using the PV detection ($PV_{NV}^{off}$ = − 810 mV). **c** ODMR spectrum from the same single NV centre as (**a**) ($PL_{NV}^{off}$ = 94.7 kcounts/s). **d** Optically detected Rabi oscillations obtained for the same single NV centre as (**b**) ($PL_{NV}^{off}$ = 44.3 kcounts/s). Both PL and PV signals were obtained simultaneously on the NV centre circled in light blue in Fig. 2. For all panels, the data (blue points) are fitted (brown line) with a Lorentzian function (magnetic resonance) and with a sinusoidal function (Rabi oscillations). Error bars represent the standard deviation (SD) of independent measurements (accumulations, see below). The experimental parameters for the magnetic resonance experiments are: number of single envelope-based (see SI) measurements per frequency point − 50, integration time per point − 3 s, number of accumulations − 10, total accumulation time − 30 min, laser power − 500 μW, MW power − − 30 dBm. The accumulation parameters for the Rabi experiments are: number of single envelope-based (see SI) measurements per frequency point − 100, integration time per point − 12 s, number of accumulations − 14, total accumulation time − 84 min, laser power − 750 μW, MW power − − 14 dBm. The KPFM detection parameters are the same as for imaging (see the caption to Fig. 2).

but from the ionization of the local trap. Another reason could be, that the charge cycling process occurs between the positively charged and the neutral state of the NV centre. To our knowledge, there is no experimental demonstration of photoemission of an NV centre in the $NV^+$ charge state, which could explain the absence of the luminescence signal. The $NV^+ \rightarrow NV^0$ transition (3.09 eV) could be explained by a two-photon process resulting in a promotion of an electron from the diamond valence band to the $NV^+$ defect[32]. However, the mechanism of the backward $NV^0 \rightarrow NV^+$ transition remains unclear. Finally, this could also be another type of defect exhibiting a charge cycling process as the one demonstrated with NV centres via the carrier-to-photon conversion method[33]. In all the cases discussed above, the PV signal from such a defect would not show the NV centre magnetic resonance, which we also did not observe experimentally.

Next, we demonstrate the readout of a single NV centre spin state through the surface photovoltage. Analogously to optically, electrically and photoelectrically detected magnetic resonance (ODMR, EDMR and PDMR, correspondingly), we call this technique Surface Voltage Detected Magnetic Resonance (SVDMR). During the measurements, the laser was constantly illuminating an NV centre and the measured KPFM signal was correlated with the applied MW frequency swept around the NV electron spin's resonance. An SVDMR spectrum obtained on a single NV centre is shown in Fig. 3a. The contrast of the SVDMR signal can be defined as

$$C_{PV}^{MR} = \frac{PV_{NV}^{on}(\nu) - PV_{NV}^{off}}{PV_{NV}^{off}} = \frac{\Delta PV^{MR}(\nu)}{PV_{NV}^{off}}, \qquad (1)$$

where $PV_{NV}^{on}(\nu)$ and $PV_{NV}^{off}$ are PV signals from the NV centre with MW turned on and off at the frequency $\nu$.

In order to be useful for quantum sensing and computing applications, the detection method should allow spin state readout for single-qubit operations. We demonstrate this by coherently driving the NV spin via Rabi oscillations and detecting its state through KPFM (Fig. 3b). Due to the long response time of the signal (on the order of 10 ms, see SI and Supplementary Fig. 5d) caused mainly by the lock-in nature of the KPFM detection and by presumably low carrier capture rate for the surface states[34], both SVDMR and Rabi experiments were conducted in a pulsed manner with MW output encoded in a low-frequency envelope similar to the one reported for PDMR on NV centres[35] (see details in Supplementary Information). The contrast of PV-detected Rabi oscillations, was estimated similarly to the SVDMR signal contrast as shown in SI. For comparison, simultaneously obtained optically detected magnetic resonance and Rabi oscillations are also shown in Fig. 3c and d, correspondingly. The contrast in optically detected magnetic resonance and Rabi experiments is defined analogously to the one defined for the experiments with PV detection using a substitution $PV \rightarrow PL$.

## Discussion

For a rigorous description of the occurring processes, two-dimensional numerical simulations of a time-dependent model based on solving Poisson's equation for the electric potential and drift-diffusion equations for the charge carriers could be used. This numerical model must include the generation of charge carriers of both types from various defects during the light illumination, as well as the trapping of defects. As such numerical simulations are out of the scope of the present work, we use instead a simplified toy model presented below for a qualitative description of the effects observed in the experiments.

The model is based on partially filled surface acceptor states originating either from $sp^2$ defects (double carbon C=C bonds)[17,36] or oxygen-terminated sites (C=O bonds)[17,25,37], which both are able to produce free holes in the diamond valence band under green laser illumination (2.38 eV). After the excitation, these holes do not freely

diffuse but rather drift in the applied AC potential $V_{AC}$ (which is used for KPFM detection) between the grounding contact and the cantilever leaving the negative charge at the excitation spot. Due to high carrier mobility in diamond, the hole's drift time is negligible compared to the $V_{AC}$'s oscillation period (see SI). Thus, one can assume, for simplicity, that during half of the oscillation period, the holes gather below the cantilever, and for the next half, they are at the metal grounding contact. This increased hole concentration leads to the trapping of charge carriers by the surface states at the cantilever tip position, resulting in a trapped positive charge at that location. A similar static model is typically used in the experiments on NV centre charge state manipulation by a cantilever-assisted local application of high electric fields[29,38].

This model allows us to qualitatively explain the PV imaging mechanism in the following way. When the laser is focused at the position of the cantilever tip, the surface states can only release holes, because the photon energy is not high enough to excite electrons to the conduction band, thus leading to an accumulation of a negative charge at the illumination spot. Therefore, even though the holes repopulate the surface states when they are next to the cantilever, the net charge trapped there is still negative, resulting in negative PV (Fig. 4a). When the laser is focused on the diamond surface away from the cantilever, the excited holes drift towards the cantilever position, leading to the positive PV (Fig. 4b). Finally, the excitation of an NV centre produces both holes from the surface states as well as electrons and holes from the NV centre itself. In this case, charge carriers of both types move towards the cantilever to be trapped there (Fig. 4c). This reduces the absolute value of the observed PV (compared to the surface illumination), but its sign depends on the carrier concentrations and capture cross-section that in turn depends on the local surface environment. Due to the spin-dependent nature of the charge carrier emission from the NV centre (see Fig. 1a, b), the associated with them PV signal is spin-dependent, which is demonstrated with the SVDMR and PV-detected Rabi experiments. Additional details of the PV mechanism can be found in the Supplementary Information.

Therefore, in this simplified qualitative picture, we assume that the charge carrier motion is governed by the drift in the applied AC potential, neglecting diffusion and trapping, which would result in a decrease of the amount of carriers reaching the cantilever. Moreover, in this consideration, we also neglect the influence of substitutional nitrogen defects, which are sources of free electrons under the laser illumination[39], and other donor-type defects, since their concentration is very low, and our measurements imply a transfer of a net positive charge when the laser is focused on the diamond surface.

From the mechanism presented above it follows that the amplitude of the AC oscillating voltage $V_{AC}$ used for KPFM detection is crucial, and the obtained images should show a strong dependence on the applied AC voltage amplitude and probably on its frequency when it becomes comparable to the carrier capture rate of the surface defects. Indeed, the amplitude dependency is observed in a series of experiments where the PV images were taken at different AC voltage amplitudes while keeping the same cantilever-contact distance (Supplementary Fig. 4a–c). In order to confirm that this effect depends on the strength of the applied electric field, we also performed measurements at different distances between the tip and the grounding contact, while keeping the $V_{AC}$ constant (Supplementary Fig. 4d–f). In both cases, the image contrast changes its sign due to the decrease in the electric field and consequent change of the carrier motion from drift to diffusion, which leads to a higher noise and to a lower absolute contrast. Though the application of a higher AC voltage amplitude could boost the signal, it also increases the attractive force between the cantilever and the grounding contact, leading to instability of the cantilever position on the sample surface.

The NV PV signal also shows a dependence on the laser power (see Supplementary Fig. 3a), where the saturation of the signal at higher

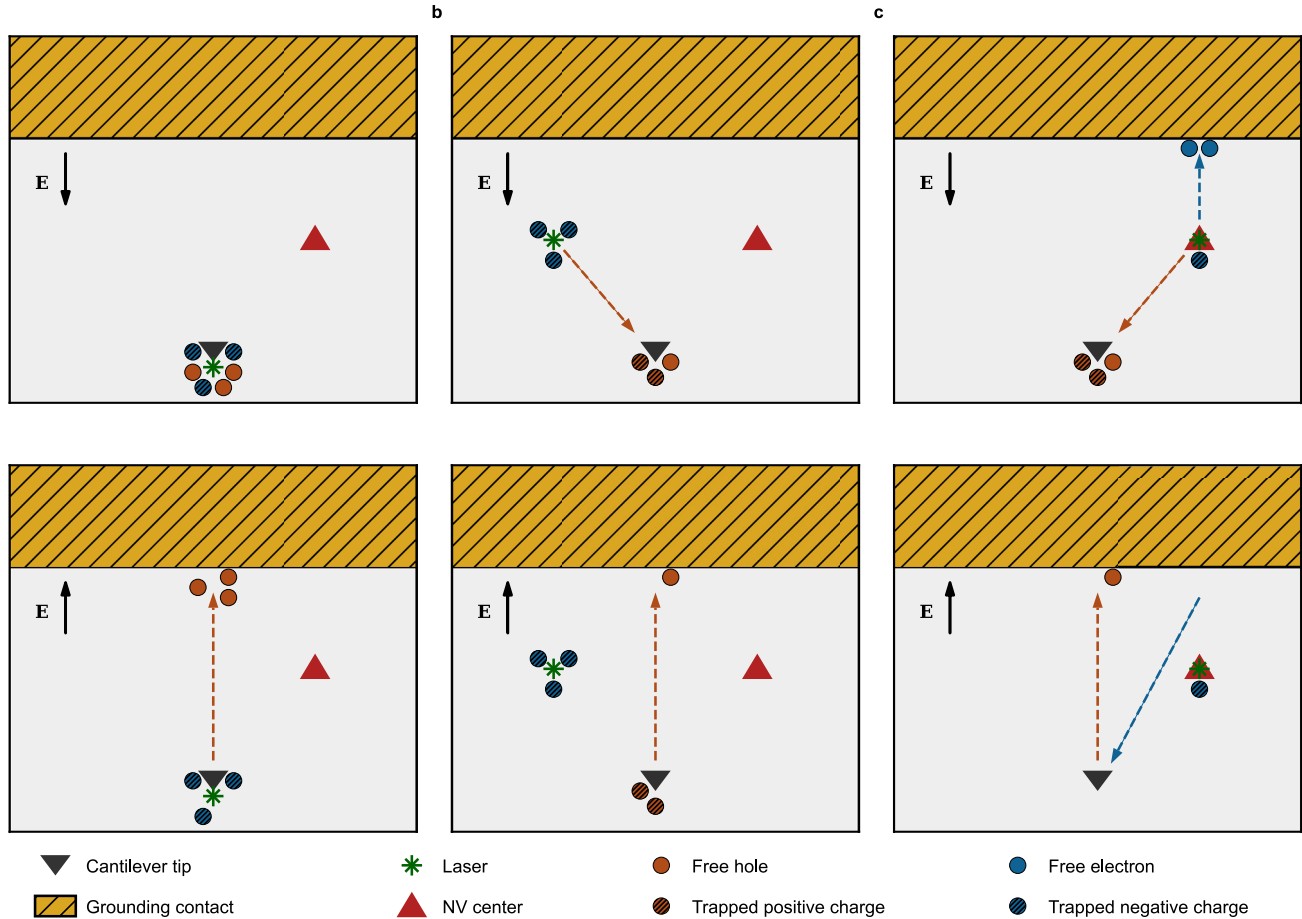

**Fig. 4 | A schematic illustration of the photovoltage origin.** All figures show a diamond surface (grey), a grounding contact (hatched gold), and a cantilever position (black triangle) viewed from the top during the application of the AC potential (black triangles serve only as an indication of the electric field direction to or from the cantilever tip) at two points in time separated by half of the oscillation period $T_{hp} = \frac{1}{2f_{AC}}$. Brown (blue) dashed arrows indicate the movement of free holes (electrons). **a** Top: the laser-focused on the cantilever excites the surface states, which release free holes. Bottom: same after $T_{hp}$. The holes drift in the applied electric field **E**, leaving the trapped negative charge unscreened. **b** Top: the laser is focused on the diamond surface away from the cantilever. The created free holes, move towards the cantilever driven by the applied $V_{AC}$. Bottom: same after $T_{hp}$. The free holes move away from the cantilever, but the trapped charges remain. **c** Top: the laser is focused on the NV centre. Free carriers are separated by the applied field, and holes populate the surface states at the tip position. Bottom: same after $T_{hp}$. The electrons from the NV centre are carried to the cantilever, decreasing the trapped positive charge. The measured PV signal, in this case, is spin-dependent (see also Fig. 1a).

powers is presumably caused by the screening of the applied $V_{AC}$ by the built-up charge and/or low density of surface states, compared to the local carrier concentration. The right choice of surface orientation and termination could raise carrier capture rates, diminishing the response time of the signal and increase the density of states enhancing the measured photovoltage.

In summary, we presented here a photovoltage-based method for spatial imaging and the readout of a single electron spin as an alternative to the existing electrical readout techniques, while the full potential of the technique is still to be studied. One of its possible advantages over the photocurrent-based readout is that it does not rely on a current flowing though the sample. This allows to simplify the design of quantum sensors, as PV-based methods do not require a low contact resistance at the sample-electrode interface. The surface photovoltage detection could also enable the implementation of fast (nanosecond scale) experimental techniques like capacitive measurements[15,40], eliminating the need for lock-in detection and thus decreasing the measurement time. Another interesting result is the detection of a non-fluorescing defect via the surface voltage, which demonstrates the potential of our technique for searching novel defect centres in diamonds. Experiments with varying the laser wavelength and power could elucidate the nature of this defect. As the measurements described here were conducted on single NV centres, the application of the photovoltage-based method to ensembles of defects must be verified experimentally. For NVs, it might be challenging, since they act as hole traps with a large capture cross-section (see, for example,[24]), thus blocking charge carriers from reaching the surface. The SVDMR technique reported in this work could be potentially applied to other promising quantum sensors like $V_2$ ($V_{Si}^-$) centres in silicon carbide[8].

## Methods
### Materials
A thin electronic grade diamond plate (3 mm × 3 mm × 50 μm) grown by the chemical vapour deposition method (CVD) by Qnami was used as a sample. Shallow single NV centres were created by nitrogen ion implantation (energy 5 keV, dose $5 \cdot 10^9$ ions/cm²) and subsequent annealing performed in the following steps[41]: 1 h ramping from room temperature to 400 °C, 4 hours at 400 °C, 1 hour ramping from 400 °C to 1000 °C, 2 h at 1000 °C, cooling down. After annealing, a tri-acid mixture (nitric, perchloric, and sulphuric acids) was used to clean the diamond surface. To apply microwaves and an additional electric potential, gold electrodes were deposited on the surface in a way similar to the one described in ref. 42 with

chromium as a sacrificial layer. A photo of the sample is shown in Supplementary Fig. 1a. The large metal pads were grounded, and the strip lines were used to apply MWs. Since the MW frequency is high, compared to the frequency of the applied AC potential, the average voltage (on timescales larger than 1 ns) applied to the strip line is 0 V, and it can be considered grounded in all experiments. The distance between the grounding and the MW contacts is 20 μm. Before the experiments, the sample was cleaned with acetone using a sonicator.

## Experimental setup

All measurements were performed on a combined Confocal-AFM setup, consisting of an AFM system (NX12, Park Systems) combined with a home-built confocal microscope[43]. A schematic of the setup is shown in Supplementary Fig. 1b. For NV excitation, a continuous wave (CW) laser diode (RLT520-80MGS, Roithner LaserTechnik) with a wavelength of 520 nm was used. Laser pulses for Rabi experiments were created using an acousto-optic modulator (AOMO 3350-199, Gooch and Housego). The laser was focused on the sample surface with an oil objective (UPLXAPO100XO, Olympus). The luminescence, collected by the same objective, was filtered with optical filters (FELH0650 and FESH0750, Thorlabs) and detected by an avalanche photodiode (SPCM-AQRH-44, Excelitas). The confocal scanning was performed using galvo mirrors (GVS012/M, Thorlabs). For magnetic resonance experiments, microwaves were generated by an MW source (APSIN 6010, Anapico) and, after amplification (ZHL-16W-43-S +, Mini-Circuits), applied to the sample via metal electrodes. For controlling the AFM and data acquisition from it, SmartScan operating software was used. Controlling of the experiment and data acquisition in the confocal part of the setup was done using Qudi software suite[44]. During KPFM measurements, the surface potential was measured using conductive cantilevers (Multi75E-G, BudgetSensors, and PPP-EFM, Nanosensors) in a frequency-modulated sideband KPFM mode[30]. The typical parameters of the AC voltage applied to the cantilever are: frequency − 3 kHz amplitude − 6 V.

## Data availability

The data used in this study are available in the Figshare database under the accession code https://doi.org/10.6084/m9.figshare.28603919. Any additional information is available from the corresponding author upon request.

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

## Acknowledgements

The authors thank Thomas Dittrich and Maxim Simmonds from Helmholtz-Zentrum Berlin and Petr Siyushev, Michael Petrov, and Milos Nesladek from Hasselt University for fruitful discussions. We are grateful to Park Systems for the experimental and financial support. This work was supported by the German Research Foundation (DFG, grants 410866378, B.N., and 410866565, B.N.) and by the German Federal Ministry of Education and Research (BMBF, grant 16KISQ034K, B.N.).

## Author contributions

S.T. conducted the measurements under supervision of B.N.; S.T. processed the raw data under supervision of B.N.; S.T. and B.N. developed a theoretical model of experiments; S.T., K.L. and B.N. wrote the manuscript and SI; K.L. and B.N. acquired funding and supervised the research process.

## Funding

## Competing interests

The authors declare no competing interests.
