## [Transparent Peer Review file · Nature Communications]

Voltage detected single spin dynamics in diamond at ambient conditions

Corresponding Author: Dr Boris Naydenov

Version 0:

Reviewer comments:

Reviewer #1

(Remarks to the Author)

Trofimov et al report a new means of electrically detecting NV centres in diamond and their magnetic resonance signatures. Charge cycling of NV centres creates free charges that are transported to surface traps near an electrically charged cantilever, which reads out a surface voltage. Since the generation of charges from the NV is spin-dependent, the surface charge accumulated is also spin dependent, and allows for quantum state readout without collecting photons or directly collecting charges as a current. A single-site emitter that is nonfluorescent but emits charges can also be found, as has been shown in PC measurements. The paper reports fascinating new results that will be of great importance to anyone working in the field of optically and electrically active defects in wide-bandgap semiconductors, and in particular those who are seeking to improve on electrical detection schemes. However, the paper lacks key details that make it difficult to fully compare against existing techniques and potentially allow the work to be reproduced. It also poses several questions that are either within the scope of the existing work or of sufficient importance to be identified and noted. I can therefore only recommend publication after these more significant points have been addressed satisfactorily.

Major points:

1. Dark signal subtraction: The authors say that their procedure scans the laser beam rather than the cantilever (I think when they say "classical" AFM on pg 3 a better term could be used), and then say that "To obtain photovoltage (PV) images characterizing changes in the sample surface potential induced by the laser illumination, we subtracted the KPFM signal measured in the dark from the CPD images" - I don't follow, were they "scanning" an extinguished laser beam around, or to obtain the KPFM image in the dark they scanned the probe?
2. Referring to the "dark" defect in Fig 2. The authors say "This "dark" defect could be an NV centre in the neutral charge state, though it did not show spin state contrast in KPFM (see below)." I have a few issues with this. Firstly, the NV signal comes from NV charge cycling creating free charges that are trapped at the surface and thus detected by the probe. So, an NV cannot be just "in the NV0 state", since it's continuously charge cycling under green light. If the authors mean that it's an NV *stuck* in the NV0 state, then it can't charge cycle and so should not create a signal. And, I don't understand the significance of it not showing spin contrast: if it were "an NV0", it would not be expected to show spin contrast. Could it perhaps be an NV+ cycling between 0 and +? Did the dark defect exhibit any different phenomenology, eg different PV power dependence, dependence on applied tip potential?
3. I think there is a somewhat oversimplified picture of the charge transport processes occurring. For example, it can be seen that as the radial distance of the NVs to the cantilever tip increases, the PV signal gets fainter. My understanding of the situation is that two things are occurring: optical excitation of the NVs creates electrons and holes, which then diffuse and drift, some towards the cantilever tip. Charges created by the NVs and then trapped by the surface states appear as a PV signal. However, the charges are subject to myriad scattering and trapping events along the path from the illumination point to the tip - they do not drift ballistically - and are also subject to complicated space charge effects (see PRL 125 256602 2020) that may be partially mitigated by the AC field, and I can only imagine are much more complicated near the surface. Nevertheless, all of this results in less charges making it to the tip and a lower PV signal, and should at least be mentioned. It would be very helpful to know the dwell time per pixel (and number of pixels) used in Fig 2, as the charge transport time can also be quite fast (in the bulk, see Physical Review B 109 (13), 134106 2024). In the SI it says only that the PV signal is measured over several periods of fAC ~1kHz.

4. I think a few basic details of the measurement in Fig 2 should be reported in the main text or caption: laser power, lock in time constant, lock in gain, scan rate, number of pixels, total scan time (the last being particularly interesting for comparison between photocurrent measurements). Which NV is being measured in Fig 3? Similarly, integration time per point should be noted for the comparison between VDMR and ODMR in Fig 3 (similar details are provided in the SI, but should also be in the main text).

5. The role of the AC potential applied to the cantilever tip should be explained properly as well (my assumption is that it is for the lock-in detection in KPFM). Potential applied to the tip, frequency, etc should be described fully. Indeed, the authors note that it is "crucial", yet do not actually mention at all what it is in the main text, except at the very end and in the supplement. Why is the electric field directed as shown in Fig 4, one would expect it would be very inhomogenous and directed radially toward the tip?

6. The authors should include reference to Nat Comms 12, 2457 (2021) and Nat. Phys. 18, 1317 (2022) which report on similar nanoscale charge engineering, and in particular pay special attention to the role of substitutional N and other complexes in the electric charge environment at the surface - which is not covered in this paper.

7. Is the PV signal in Fig S2e quadratic in laser power, as would be expected in a 2-photon mediated ionisation/recombination process? What is the role of nitrogen - photoionisation of N should be significant given it will be more abundant than the NVs and generate electrons.

8. The authors refer to a 'model' that describes their experiment, but it's more of what I would call a mathematically justified hand-wave. I'm not sure why the description is split 50/50 between main text and SI when the SI includes a more tractable and rigorous explanation. A model would numerically reproduce the key results, ideally based on some sort of drift-diffusion-poisson equation framework. I realise that this is a complicated and laborious undertaking outside the scope of the present work (which shows quite conclusively that they can detect NV centres with KPFM), but an overview of what would be required to more rigorously justify the underlying theory would be appreciated.

9. In the (very brief) conclusion, the authors say that an advantage of the technique over other electrical detection methods is that it does not detect current. I agree, but it would seem some of the necessary evils of a current measurement carry over (like strong electric fields of order volts per micron, and lock in amplifiers with slow time constants) and a few additional complexities such as near surface operation and some ambiguities when deducing the sources of charge/current, and the latter is of significant concern. There are details missing from the paper that make it a bit hard to independently determine if PV is indeed better than PC detection (such as measurement durations, the role of N and so on). I think the authors should more comprehensively address this in the conclusion/discussion: what are the most important advantages of surface voltage readout vs photocurrent? Are some of the problems in photocurrent measurements more tractable when ported to PV? Does the technique work for deeper NVs, and how shallow does an NV need to be to work? How could it be adapted to identify nonfluorescent defects?

Minor points:

1. When referring to ref 22, the authors should clarify that this is a hole capture process: NV charge cycling and thus hole generation is spin dependent, and SiV⁻²⁻ is a hole trap, not an electron trap.

2. Fig 1: It does not make sense to show multi-electron states (3A2, 3E etc) and the CB and VB of the diamond in the same level diagram. I am aware that the diagram is purely illustrative, since no energy splittings are shown (and nor can they be accurately represented in such a diagram) but the relative energy splittings could be misinterpreted. See Phys. Rev. B 104, 235301 for more details.

3. It might pay to indicate in the caption of Fig 2 or the figure itself exactly where the probe cantilever is (I'm assuming the blue mess in the centre). Similarly, the bright strip at the top should also be defined/explained (mw/gnd electrode?).

4. I think Fig S3 would be more informative if the colour scales were the same, i.e. it would be easier to see that the signal altogether reduces and not that the background increases dramatically, which is how it appears. The labels for electric field strength (3.0V/um etc) are not defined: where is this defined, at the tip? Potential should be used instead.

5. While I see the rationale for "V"DMR, voltage detected magnetic resonance, it belies the intricacy of the technique. Something like "surface charge detected MR" would be closer to reality.

Reviewer #2

(Remarks to the Author)

In the paper Voltage detected single spin dynamics in diamond at ambient conditions the authors report the detection of the spin resonance of a single NV center by Kelvin probe force microscopy (KPFM). Measuring the diamond surface photovoltage under sub-band (green) illumination the authors demonstrate the detection of the NV center spin resonance spectrum and the spin Rabi oscillations. The NV center detection and its spin readout using this technique are qualitatively explained in the text. Under green illumination, free holes are produced in the diamond valence band by partially filled surface acceptor. Due to the AC potential applied by the cantilever, holes can drift and be trapped close to the cantilever generating positive charges that induce a positive surface potential. The NV center, under illumination, generates electron-hole pairs. Since at the laser power used in the experiment the surface potential is saturated (it does not change increasing the hole concentration), the main consequence of the NV center excitation is the drift of electrons close to the cantilever and thus the reduction of surface potential. Since the electron-hole generation rate of the NV center is spin dependent, the NV center spin state can be readout looking at the surface potential fluctuation.

The paper is interesting, very clear and easy to read. The main novelty here proposed is the spin readout by KPFM. The methodology, the model and the results achieved demonstrate a worthy and pioneering work that deserves the publication on Nature Communication after a detailed comparison between the advantages of the proposed technique to existing techniques (ODMR and PDMR) in terms of both technical implementation and performance.

Here I detail some minor concerns about the paper.

- Reference 5 is about a comparison between EPR and EDMR and not between EDMR and ODMR as it would be expected by the context.
- Page 4, eq 1. Why are you defining the contrast using the absolute values at the denominator?
- Page 4, line 5. In bracket you write (in absolute values) but the value you give is negative (-57%). What does it mean in absolute values?
- Page 4, eq2. Why are you using PV_on and PV_off and not PV_NV_On and PV_NV_Off? Even if it is the same, it will be more familiar to ODMR community. Moreover, the plot in fig3a will be easier to read with PV_NV instead of PV, since it will allow an easier comparison between the voltage on the left y axis and the contrast on the right y axis. In fact, in the way you proposed, the value of PV_NV must be found from fig2c and it is not evident. In case your choice of using PV_on and PV_off is justified by some arguments, it could be useful to report the value of PV_NV_Off in the caption of the figure 3a/b.

- Page 5, fig3a. See previous comment.
- Page 5, fig3c-d. How do you define the ODMR contrast? Why is it negative?
- Page 5, line10. The positive contrast of VDMR (and, therefore, Rabi oscillations) is a consequence of a negative PV_NV signal from an NV centre.

I would say it is more a consequence of how you defined the contrast using the absolute value of PV_NV. Why do you use an absolute value at the denominator of eq. 2? Why do you need this sentence here? To compare it to the ODMR contrast?

- Page 6, line1: In ref [17] there are also sources of electrons cited for oxidized surface like yours. Do the experimental data allow you to neglect it or the sample fabrication procedure? Moreover, what about nitrogen in your sample. Does it play a role?
- Page6, line 23. You explain the detection of the NV center but not its spin readout. You do that in the SI. I suggest to move that explanation in the main text.
- Page 6, line 29. Probably on its frequency. Have you done some experiments that you can put in SI? Why may you expect a dependence on the frequency? Is it related to the NV center mobility?
- Page 6, line 42-43. Probably this sentence is better in the previous paragraph, when you discuss about AC amplitude.
- Page 7, line4-5. Comments more on the advantages of this technique for NV centers or other defects. You state essentially two advantages:

1) Simplification of the design. Will the presence of a cantilever be a problem in the design of the system and its use? Do you necessarily need it?

2) Implementation of fast nanoscale technique. Considering the KHz frequency of the cantilever, how can you measure nanosecond effects?

Moreover, the results you obtained, when compared to ODMR, are not so impressive (Contrast, SNR). This is understandable as it is a first and successful experiment of a new spin detection technique. But, can you estimate how much you could improve that? Do you expect results comparable to ODMR or PDMR? Can your technique be more effective than PDMR for some kind of defects in some materials? In that case, what are the specific characteristics of the defects and the surfaces that allow your technique to perform better than PDMR? Could you use this technique for ensembles?

Here I detail some minor concerns about the SI.

- SI2. Line 6. Why do you use that formula for fitting PV_surf?
- Fig S2 f. What do you mean by saturation in this plot? How can you say that the saturation is above 20 μ w (line 23, SI3)
- Fig S4c. Is the envelope and so the lock-in present also when MW are off?
- Fig S4f: Do you find any dependence of the contrast and shape of the Rabi oscillations on laser power and pulse duration? How did you choose the laser power and pulse duration?

Version 1:

Reviewer comments:

Reviewer #1

(Remarks to the Author)

I am happy that the authors have responded satisfactorily to my comments and the paper can be published in Nat Comms. It's a good paper that introduces an interesting new probe to help understand some very vexing problems in diamond. One minor point to make is towards the author's citation of Ref 2 (Peruzzo paper 2021) regarding NV+ PL: This paper is a bit of an outlier, I think it's now pretty well established NV+ is optically dark. At least no one has experimentally reported NV+ PL that I am aware of.

Reviewer #2

(Remarks to the Author)

The authors answered my questions and edited the manuscript accordingly.

I have four minor comments that may perhaps help the reader to better understand the paper:

- 1) In the main text, you define the imaging contrast (eq1) to evaluate the visibility of the NV center with respect to the background, but you do not use this parameter elsewhere in the text. Then you define in eq2 the magnetic resonance contrast that you use for quantitative measurements all along the text. I will suggest, but the choice is up to you, not to call "imaging contrast" the parameter of eq1 and so to use a different letter for this parameter. I think that can make easier the reading of the paper.
- 2) Page 8: "The NV PV signal also shows a dependence on the laser power (see Fig.:S2e), where the saturation is presumably caused by the screening of the applied VAC by the built-up charge and/or low density of surface states, compared to the local carrier concentration."
Without reading the supplementary material, I think the reader cannot understand what saturation you are talking about. Maybe you can rephrase a little bit this sentence introducing the fact that the NV PV signal saturates increasing the laser power.
- 3) In the SI, S2, it is not clear if the fitting model for PV surface is based on an expected linear behaviour or, after having tested different models, the linear is the one that best fit your results. It could be interesting to add an image showing the fit, as you do for PV NV in figS2c.
- 4) Figure S4a-b, you compare the CW and the pulsed acquisition procedure. It would be better, in my opinion, to use the same parameter on the Y-axis to make the comparison.

Voltage detected single spin dynamics in diamond at ambient conditions

Sergei Trofimov, Klaus Lips and Boris Naydenov

January 27, 2025

Response Letter to Reviewers

1 Response to Reviewer #1

1.1 Comments

Comment 1. Dark signal subtraction: The authors say that their procedure scans the laser beam rather than the cantilever (I think when they say “classical” AFM on pg 3 a better term could be used), and then say that “To obtain photovoltage (PV) images characterizing changes in the sample surface potential induced by the laser illumination, we subtracted the KPFM signal measured in the dark from the CPD images” — I don’t follow, were they “scanning” an extinguished laser beam around, or to obtain the KPFM image in the dark they scanned the probe?

Response 1. We agree with Reviewer 1, that the description of this part of the experiment could lead to confusion. Since we measure KPFM signal effectively in one point on the sample surface (laser scanning technique), as the dark signal we use the KPFM signal measured in the same spot in the dark before turning on the laser used for scanning.

In order to make this clearer to readers, we changed the sentence “To obtain photovoltage (PV) images characterizing changes in the sample surface potential induced by the laser illumination, we subtracted the KPFM signal measured in the dark from the CPD images” on page 3 to “To obtain photovoltage (PV) images characterizing changes in the sample surface potential induced by the laser illumination, we subtracted from the CPD images the KPFM signal measured beforehand with the laser turned off”.

We have also changed the expression “classical AFM scanning technique” to “probe scanning technique”.

Comment 2. Referring to the “dark” defect in Fig 2. The authors say “This ‘dark’ defect could be an NV centre in the neutral charge state, though it did not show spin state contrast in KPFM (see below).”. I have a few issues with this. Firstly, the NV signal comes from NV charge cycling creating free charges that are trapped at the surface and thus detected by the probe. So, an NV cannot be just “in the NV⁰ state”, since it’s continuously charge cycling under green light. If the authors mean that it’s an NV *stuck* in the NV⁰ state, then it can’t charge cycle and so should not create a signal. And, I don’t understand the

significance of it not showing spin contrast: if it were “an NV^0 ”, it would not be expected to show spin contrast. Could it perhaps be an NV^+ cycling between 0 and +? Did the dark defect exhibit any different phenomenology, e.g. different PV power dependence, dependence on applied tip potential?

Response 2. The reviewer correctly points out that, the PV signal comes from the charge cycling process. Since we observe a PV signal from this particular defect, we conclude, that there is a charge cycling process going on. In our experiment the PL signal originates only from NV^- and not NV^0 as the detection window for the fluorescence is in the wavelength region of 650–750 nm. So it could indeed be, that this dark defect could be NV^+ . We have included the following text in the revised manuscript (Results section), which elaborates more on this observation.

The absence of luminescence could indicate a presence of an NV centre with a proximal local electron trap such as nitrogen centres, vacancy complex, or other acceptor states (see, for example, [1]). In this case, the NV centre spends more time in the neutral charge state during illumination. The negative PV signal then would originate not from the $NV^- \rightarrow NV^0$ transition itself, but from ionization of the local trap. Another reason could be, that the charge cycling process occurs between the positively charged and the neutral state of the NV centre. As shown by *ab-initio* calculations in [2], the luminescence spectrum of a positively charged NV centre is below 1.65 eV in photon energy, or above 775 nm in photon wavelength, therefore out of our detection range 650–750 nm, which could explain the absence of the luminescence signal. The $NV^+ \rightarrow NV^0$ transition (3.09 eV) could be explained by a two photon process resulting in a promotion of an electron from the diamond valence band to the NV^+ defect. However, the mechanism of the backward $NV^0 \rightarrow NV^+$ transition remains unclear. Finally, this could also be another type of defect exhibiting charge cycling process as the one demonstrated with NV centres via carrier-to-photon conversion method [3]. In all the cases discussed above, the PV signal from such a defect would not show the NV centre magnetic resonance, which we also did not observe experimentally.

Comment 3. I think there is a somewhat oversimplified picture of the charge transport processes occurring. For example, it can be seen that as the radial distance of the NVs to the cantilever tip increases, the PV signal gets fainter. My understanding of the situation is that two things are occurring: optical excitation of the NVs creates electrons and holes, which then diffuse and drift, some towards the cantilever tip. Charges created by the NVs and then trapped by the surface states appear as a PV signal. However, the charges are subject to myriad scattering and trapping events along the path from the illumination point to the tip - they do not drift ballistically - and are also subject to complicated space charge effects (see PRL 125 256602 2020) that may be partially mitigated by the AC field, and I can only imagine are much more complicated near the surface. Nevertheless, all of this results in less charges making it to the tip and a lower PV signal, and should at least be mentioned. It would be very helpful to know the dwell time per pixel (and number of pixels) used in Fig 2, as the charge transport time can also be quite fast (in the bulk, see Physical Review B 109 (13), 134106 2024). In the SI it says only that the PV signal is measured over several periods of $f_{AC} \approx 1$ kHz.

Response 3. We agree with the reviewer, that the occurring processes are more complicated than what we described in the manuscript. Indeed, there is not only a drift of the charge carriers, but also other processes like diffusion, scattering, and trapping should be considered. We used a simplified description to be able to qualitatively explain the observed effects. In the Supplementary Information (SI) we show, that an increase in the alternating

current (AC) voltage amplitude leads to a stronger PV signal and higher imaging contrast. Therefore, we conclude that while the other processes are undoubtedly present, the observed effects are mainly governed by the drift in the applied AC potential under these experimental conditions. It is important to mention the other processes affecting the charge carrier motion during the experiment, therefore, we have included the following text in the manuscript (Discussion section).

Therefore, in this simplified qualitative picture we assume that the charge carrier motion is governed by the drift in the applied AC potential, neglecting diffusion and trapping, which would result in a decrease of the amount of carriers reaching the cantilever. Moreover, in this consideration we also neglect the influence of substitutional nitrogen defects, which are a source of free electrons under the laser illumination [4], and other donor-type defects, since their concentration is very low, and our measurements imply a transfer of a net positive charge when the laser is focused on the diamond surface.

We have also included the missing information about the signal accumulation to the caption of Fig. 2:

The accumulation parameters are: image size — 150 x 150 pixels, scanning rate — 200 Hz (5 ms per pixel), total scan time — 112.5 s, laser power — 500 μ W. The KPFM detection parameters are: AC potential amplitude V_{AC} — 6 V, AC potential frequency f_{AC} — 3 kHz, lock-in time constant — 1 ms, lock-in sensitivity — 1 V, voltage feedback loop gain — 0.5.

Comment 4. I think a few basic details of the measurement in Fig 2 should be reported in the main text or caption: laser power, lock in time constant, lock in gain, scan rate, number of pixels, total scan time (the last being particularly interesting for comparison between photocurrent measurements). Which NV is being measured in Fig 3? Similarly, integration time per point should be noted for the comparison between VDMR and ODMR in Fig 3 (similar details are provided in the SI, but should also be in the main text).

Response 4. Indeed, the experimental parameters are crucial for comparison to the existing imaging techniques and for reproducing the results. We have included the missing details to the caption of Fig. 2 (see also Response 3).

We have marked in Fig. 2 the NV centre that was measured in Fig. 3.

We have also provided the measurement parameters for the magnetic resonance and Rabi experiments in the caption of Fig. 3: The experimental parameters for the magnetic resonance experiments are: number of single envelope-based (see SI) measurements per frequency point — 50, integration time per point — 3 s, number of accumulations — 10, total accumulation time — 30 min, laser power — 500 μ W, MW power — -30 dBm. The accumulation parameters for the Rabi experiments are: number of single envelope-based (see SI) measurements per frequency point — 100, integration time per point — 12 s, number of accumulations — 14, total accumulation time — 84 min, laser power — 750 μ W, MW power — -14 dBm. The KPFM detection parameters are the same as for imaging (see the caption to Fig. 2).

Comment 5. The role of the AC potential applied to the cantilever tip should be explained properly as well (my assumption is that it is for the lock-in detection in KPFM). Potential applied to the tip, frequency, etc should be described fully. Indeed, the authors note that it is “crucial”, yet do not actually mention at all what it is in the main text, except at the very end and in the supplement. Why is the electric field directed as shown in Fig 4, one would expect it would be very inhomogeneous and directed radially toward the tip?

Response 5. We thank the reviewer for pointing this out. Indeed, the applied AC

potential is needed for the lock-in KPFM detection. To make it clear to readers, we have added the following information to the main text (Introduction section):

In frequency modulated KPFM experiments [5], a probe oscillates at a frequency f_0 in the vicinity of the electrically grounded sample surface. An alternating current (AC) potential with an amplitude V_{AC} at a frequency f_{AC} applied to the probe induces mechanical oscillations at sideband frequencies $f_0 \pm f_{AC}$. The amplitude of these sideband oscillations is proportional to the contact potential difference (CPD) between the probe (cantilever) and the sample, and is detected using a lock-in technique. A voltage feedback loop is used to apply a direct current (DC) potential to the probe in order to nullify the sideband oscillations, thus measuring the CPD that reflects the electrical potential of the surface.

As the reviewer mentions, the electric field in the considered description would be very inhomogeneous. Therefore, in Fig. 4 we indicate only the direction of the electric field to or from the cantilever, not to overload the figures. For clarification, we have added the following information to the caption of Fig. 4: black arrows serve only as an indication of the electric field direction to or from the cantilever tip.

Comment 6. The authors should include reference to Nat Comms 12, 2457 (2021) and Nat. Phys. 18, 1317 (2022) which report on similar nanoscale charge engineering, and in particular pay special attention to the role of substitutional N and other complexes in the electric charge environment at the surface — which is not covered in this paper.

Response 6. We agree, that the effects reported in the suggested papers should be mentioned in our work, and they are added in the revised manuscript, as they deal with the local charge manipulation using an electrical potential applied to a conductive AFM cantilever. Therefore, we have added the following sentences to the manuscript (Introduction and Discussion sections):

Charges, trapped in these states, change the electrical potential of the surface [6] and can be indirectly probed...

A similar static model is typically used in the experiments on NV centre charge state manipulation by a cantilever-assisted local application of high electric fields [6, 7].

Indeed, we do not discuss the effect of substitutional nitrogen, which provides free electrons in the diamond conduction band under the green laser excitation. The reason for that is the following: when the laser is focused on the diamond surface, we see a positive charge that is trapped at the cantilever position. Therefore, for a simplified description of the observed effect we neglect the presence of free electrons from the nitrogen impurities since their effect is not visible due to high effect of free holes, produced by the surface states. Also, the concentration of P1 centres in our diamond is below 5 pbb ($\sim 2 \times 10^{14}$ N/cm³), according to the provider Qnami. In the implanted region it is higher, but still only $\sim 5 \times 10^{15}$ N/cm³. For a rigorous description, of course, it is important to include the nitrogen defects in the consideration. To mention it, we have added the following sentence to the main text (Discussion section):

Moreover, in this consideration we also neglect the influence of substitutional nitrogen defects, which are a source of free electrons under the laser illumination [4], and other donor-type defects, since their concentration is very low, and our measurements imply a transfer of a net positive charge when the laser is focused on the diamond surface.

Comment 7. Is the PV signal in Fig. S2e quadratic in laser power, as would be expected in a 2-photon mediated ionisation/recombination process? What is the role of nitrogen —

photoionisation of N should be significant given it will be more abundant than the NVs and generate electrons.

Response 7. Indeed, one would expect a quadratic increase of the NV PV signal from the laser power. However, as we demonstrate in Fig. S2e, this signal shows a saturation behaviour and is saturated already near 100 mW. Therefore, it was not possible to fit it with a quadratic function. As it is important to mention it, we have included the following text in the caption of Fig. S2 in SI:

The PV_{NV} signal is expected to be quadratic in the laser power due to the two-photon excitation processes, but instead we observe a saturation behaviour.

Since the nitrogen is indeed present in the sample due to ion implantation process and is more abundant than NV centres, its PV signal should be visible when the laser is focused on the diamond surface without illuminating NV centres. However, we observe a positive charge that is collected under the cantilever tip. From this we conclude, that the negative PV signal from the substitutional nitrogen is lower than the positive signal from the surface states, and neglect its effects in our considerations. To make it clearer to readers, we included a sentence about nitrogen in the Discussion section (see Response 6).

Comment 8. The authors refer to a 'model' that describes their experiment, but it's more of what I would call a mathematically justified hand-wave. I'm not sure why the description is split 50/50 between main text and SI when the SI includes a more tractable and rigorous explanation. A model would numerically reproduce the key results, ideally based on some sort of drift-diffusion-poisson equation framework. I realise that this is a complicated and laborious undertaking outside the scope of the present work (which shows quite conclusively that they can detect NV centres with KPFM), but an overview of what would be required to more rigorously justify the underlying theory would be appreciated.

Response 8. We agree with the reviewer, that this is only a qualitative description of the observed effects. As the reviewer pointed out, in our case the studied system is complicated, and such numerical simulations would be the goal of a separate study. Instead, we tried to concentrate on the effects that we observe experimentally. This is the reason why we put most of this toy model in the SI. In the main text we kept only the key points of it to be able to suggest an explanation for the observed effects. However, to improve the explanation further, we have now included the following sentence to Discussion section:

Due to the spin-dependent nature of the charge carrier emission from the NV centre (see Fig. 1 a,b), the associated with them PV signal is spin-dependent, which is demonstrated with the SVDMR and PV-detected Rabi experiments.

We also agree that a discussion of what needs to be done for numerical simulations of the experiment is important. Therefore, we have included the following text in the Discussion section:

For a rigorous description of the occurring processes, two-dimensional numerical simulations of a time-dependent model based on solving Poisson's equation for the electric potential and drift-diffusion equations for the charge carriers could be used. This numerical model must include generation of charge carriers of both types from various defects during the light illumination, as well as trapping of defects. As such numerical simulations are out of scope of the present work, we use instead a simplified toy model presented below for a qualitative description of the effects observed in the experiments.

Comment 9. In the (very brief) conclusion, the authors say that an advantage of the

technique over other electrical detection methods is that it does not detect current. I agree, but it would seem some of the necessary evils of a current measurement carry over (like strong electric fields of order volts per micron, and lock in amplifiers with slow time constants) and a few additional complexities such as near surface operation and some ambiguities when deducing the sources of charge/current, and the latter is of significant concern. There are details missing from the paper that make it a bit hard to independently determine if PV is indeed better than PC detection (such as measurement durations, the role of N and so on). I think the authors should more comprehensively address this in the conclusion/discussion: what are the most important advantages of surface voltage readout vs photocurrent? Are some of the problems in photocurrent measurements more tractable when ported to PV? Does the technique work for deeper NVs, and how shallow does an NV need to be to work? How could it be adapted to identify nonfluorescent defects?

Response 9. We thank the reviewer for this comment. The second reviewer has also raised similar points. It is still to be investigated in detail if SVDMMR has significant advantages over PDMMR, since we report here the first observation of this effect. For example PDMMR performs well for NVs few micrometres below the diamond surfaces, but this still has to be verified for SVDMMR.

We see one of the advantages of the photovoltage-based detection over the photocurrent-based one in the fact, that PV measurements do not require a low contact resistance at the sample-electrode interface, so the quality of the surface electrodes is not as important.

For the identification of non-fluorescent defects, like the one described in the manuscript, we would need to perform additional measurements varying the experimental conditions (laser power, laser wavelength) as well as apply other optical and surface characterisation methods.

We have rewritten the conclusion of the manuscript stressing the possible advantages of the PV-based detection technique over the PC-based one.

1.2 Minor points

Point 1. When referring to ref 22, the authors should clarify that this is a hole capture process: NV charge cycling and thus hole generation is spin dependent, and $\text{SiV}^{-/2-}$ is a hole trap, not an electron trap.

Response 1. We agree with the reviewer, that the process, reported in the cited paper was in later works (see, for example, [8]) assigned to a SiV^- formation from SiV^{2-} by capturing a hole. To clarify this, we have implemented corrections to the mentioned sentence:

This method was successfully demonstrated for ensembles of NV (spin-dependent hole source) and SiV (hole trap [8]) centres in diamond [9] as well as ...

Point 2. Fig 1: It does not make sense to show multi-electron states (3A2, 3E etc) and the CB and VB of the diamond in the same level diagram. I am aware that the diagram is purely illustrative, since no energy splittings are shown (and nor can they be accurately represented in such a diagram) but the relative energy splittings could be misinterpreted. See Phys. Rev. B 104, 235301 for more details.

Response 2. Indeed, the diagram is purely illustrative, though the reviewer is right, and it can be confusing. To avoid ambiguities, we have corrected Fig. 1. We have also added the mentioned reference to the manuscript.

Point 3. It might pay to indicate in the caption of Fig 2 or the figure itself exactly where the probe cantilever is (I'm assuming the blue mess in the centre). Similarly, the bright strip at the top should also be defined/explained (mw/gnd electrode?).

Response 3. We agree, that it is useful for readers to understand where the cantilever and the grounding electrode are in the demonstrated scans. Therefore, we have marked them in Fig. 2.

Point 4. I think Fig S3 would be more informative if the colour scales were the same, i.e. it would be easier to see that the signal altogether reduces and not that the background increases dramatically, which is how it appears. The labels for electric field strength (3.0V/ μm etc) are not defined: where is this defined, at the tip? Potential should be used instead.

Response 4. We agree with this suggestion and have changed the colour scales in Fig. S3.

The reviewer is indeed right, these labels are confusing. We have changed them to potential and the distance between the tip and the grounding electrode, as these values were varied in the experiment.

Point 5. While I see the rationale for "V"DMR, voltage detected magnetic resonance, it belies the intricacy of the technique. Something like "surface charge detected MR" would be closer to reality.

Response 5. We agree with the reviewer, that naming our technique "voltage detected magnetic resonance" is a bit too general, so we decided to change the name to "Surface Voltage Detected Magnetic Resonance" (SVDMR), which is indeed closer to reality.

2 Response to Reviewer #2

2.1 Comments to the main text

Comment 1. Reference 5 is about a comparison between EPR and EDMR and not between EDMR and ODMR as it would be expected by the context.

Response 1. Indeed, the reference 5 compares EPR and EDMR. We have changed it in the revised manuscript to [10].

Comment 2. Page 4, eq 1. Why are you defining the contrast using the absolute values at the denominator?

Response 2. We define the contrast using the absolute values in the denominator, because the PV_{NV} signal is negative, while the PV_{surf} is positive. Without using the absolute values, their sum would be lower than PV_{NV} , leading to a contrast higher than 100 %. Thus, we decided to use the absolute values in our contrast definition. To make it more clear to readers, we have added the following text next to the Equation 1:

In this equation we use absolute values in the denominator, since the PV_{NV} and PV_{surf} signals have different signs.

Comment 3. Page 4, line 5. In bracket you write (in absolute values) but the value you give is negative (-57 %). What does it means in absolute values?

Response 3. Indeed, this sentence is not very clear. We have corrected it: ...and calculate the maximum contrast for this image to be -57 %.

Comment 4. Page 4, eq 2. Why are you using PV_{on} and PV_{off} and not $PV_{NV_{on}}$ and $PV_{NV_{off}}$? Even if it is the same, it will be more familiar to ODMR community. Moreover, the plot in fig3a will be easier to read with PV_{NV} instead of PV, since it will allow an easier comparison between the voltage on the left y axis and the contrast on the right y axis. In fact, in the way you proposed, the value of PV_{NV} must be found from fig2c and it is not evident. In case your choice of using PV_{on} and PV_{off} is justified by some arguments, it could be useful to report the value of $PV_{NV_{off}}$ in the caption of the figure 3a/b.

Response 4. We agree that these notations can be misleading. The logic of our definition is the following. We started with the standard definition as the one suggested by the reviewer:

$$C_{PV}^{MR} = \frac{PV_{NV}^{on}(\nu) - PV_{NV}^{off}}{PV_{NV}^{off}}$$

Using the previously defined NV signal $PV_{NV} = PV - PV_{surf}$ we can rewrite the previous equation as:

$$C_{PV}^{MR} = \frac{PV^{on}(\nu) - PV_{surf}^{on}(\nu) - PV^{off} + PV_{surf}^{off}}{PV_{NV}^{off}} = \frac{PV^{on}(\nu) - PV^{off}}{PV_{NV}^{off}} = \frac{\Delta PV^{MR}(\nu)}{PV_{NV}},$$

since the background signal does not change due to microwave application. Since PV_{NV}^{off} is a PV signal from the NV centre in the absence of microwaves, it is equal to the $PV_{NV}(x_i, y_i)$ defined previously, where x_i and y_i are the position of the NV centre chosen for the magnetic resonance measurement. This value was plotted as a 2D map in Fig. 2c. For example, for the NV centre chosen for the magnetic resonance experiments (circled in light blue in Fig. 2) $PV_{NV} = -610$ mV. As the reviewer suggested, we have also put this value together with $PL_{NV} = 94.7$ kcounts/s in the caption of Fig. 3.

Considering all written above, we come to the definition that we used in the manuscript. The reason that we originally put this definition in the manuscript was simply because with our measurement technique (the microwave envelope modulation) we directly measure $PV^{on}(\nu)$ and PV^{off} (Fig. S4d) and not the $PV_{NV}^{on}(\nu)$ and PV_{NV}^{off} . Note, that here we already implemented the reviewer's suggestion from Comment 7 and do not use absolute values.

However, as the reviewer pointed out, this might be confusing for the ODMR community, so we have decided to place the standard definition of the contrast in the main text and show how it transforms to our definition in the SI:

Using that $PV_{NV} = PV - PV_{surf}$, the fact that the PV_{surf} does not depend on the MW application, and that PV_{NV}^{off} is simply PV_{NV} obtained on the measured NV centre, the magnetic resonance contrast defined in Equation 2 of the main text can be rewritten in the following way:

$$C_{PV}^{MR} = \frac{PV^{on}(\nu) - PV_{surf}^{on}(\nu) - PV^{off} + PV_{surf}^{off}}{PV_{NV}^{off}} = \frac{PV^{on}(\nu) - PV^{off}}{PV_{NV}} = \frac{\Delta PV^{MR}(\nu)}{PV_{NV}}$$

As the reviewer suggested, we have also changed the axis labels, so that they represent the physical quantity that we measure — ΔPV^{MR} .

Comment 5. Page 5, fig 3a. See previous comment.

Response 5. We have corrected Fig. 3 and its caption to make it more clear (see Response 4 and Response 6).

Comment 6. Page 5, fig 3c-d. How do you define the ODMR contrast? Why is it negative?

Response 6. We define the ODMR contrast in the same way as the contrast in PV:

$$C_{PL}^{MR} = \frac{PL_{NV}^{on}(\nu) - PL_{NV}^{off}}{PL_{NV}^{off}}$$

To make sure, that it is clear to readers, we have added the following sentence to the main text (Results section):

The contrast in optically detected magnetic resonance and Rabi experiments is defined analogously to the one defined for the experiments with PV detection using a substitution $PV \rightarrow PL$.

The ODMR contrast is negative, because it reflects the fact, that under the magnetic resonance conditions the PL signal from the NV decreases.

Comment 7. Page 5, line 10. The positive contrast of VDMR (and, therefore, Rabi oscillations) is a consequence of a negative PV_{NV} signal from an NV centre. I would say it is more a consequence of how you defined the contrast using the absolute value of PV_{NV} . Why do you use an absolute value at the denominator of eq. 2? Why do you need this sentence here? To compare it to the ODMR contrast?

Response 7. We thank the reviewer for the important comment. The mentioned positive contrast in SVDMR is indeed a consequence of the contrast definition. With the mentioned sentence we wanted to underline, that although the plotted contrast is positive, the PV signal from an NV centre still decreases under the resonance conditions, similar to ODMR (PL reduction) and typically observed PDMR (photocurrent reduction) as opposed to some special cases with PDMR on samples with higher amount of acceptor defects [10], where the PDMR contrast becomes positive, meaning that the photocurrent increases at resonance.

To make our statements clearer we have decided to remove the modulus from the definitions for the SVDMR and PV-detect Rabi contrasts, and the sentence mentioned by the reviewer. Thus, the contrast is negative for both ODMR and SVDMR, which reflects that the signal (PL_{NV} and PV_{NV} , respectively) decreases under at resonance.

Comment 8. Page 6, line 1: In ref [17] there are also sources of electrons cited for oxidized surface like yours. Do the experimental data allow you to neglect it or the sample fabrication procedure? Moreover, what about nitrogen in your sample. Does it play a role?

Response 8. There are indeed sources for electron excitation to the conduction band at the oxygenated surface. Moreover, as was pointed out by both reviewers, the sample contains nitrogen which acts as a deep donor and provides electrons in the conduction band under the laser illumination. However, we neglect this effect in our qualitative description of the observed effects, because we see a positive PV signal, when the laser is not illuminating an NV centre, which implies, that photogenerated holes are the majority charge carrier.

We have added the following text to Discussion section to clarify this:

Moreover, in this consideration we also neglect the influence of substitutional nitrogen defects, which are sources of free electrons under the laser illumination [4], and other donor-type defects, since their concentration is very low, and our measurements imply a transfer of a net positive charge when the laser is focused on the diamond surface.

Comment 9. Page 6, line 23. You explain the detection of the NV center but not its spin readout. You do that in the SI. I suggest to move that explanation in the main text.

Response 9. As the reviewer suggested, we have included the following sentence to Discussion section:

Due to the spin-dependent nature of the charge carrier emission from the NV centre (see Fig. 1 a,b), the associated with them PV signal is spin-dependent, which is demonstrated with the SVDMR and PV-detected Rabi experiments.

Comment 10. Page 6, line 29. Probably on its frequency. Have you done some experiments that you can put in SI? Why may you expect a dependence on the frequency? Is it related to the NV center mobility?

Response 10. We have not done experiments with different frequencies f_{AC} , because in the sideband KPFM measurements that we use for detection, it is possible to change the frequency only in the small range from 2 to 5 kHz. The frequency dependence may arise at high frequencies that are comparable to the carrier capture rate of the surface defects.

To clarify it in the manuscript, we have added the following part to the sentence mentioned by the reviewer:

...frequency, when it becomes comparable to the carrier capture rate of the surface defects.

Comment 11. Page 6, line 42-43. Probably this sentence is better in the previous paragraph, when you discuss about AC amplitude.

Response 11. We agree with this point and have moved the mentioned sentence to the previous paragraph.

Comment 12. Page 7, line4-5. Comments more on the advantages of this technique for NV centers or other defects. You state essentially two advantages:

1) Simplification of the design. Will the presence of a cantilever be a problem in the design of the system and its use? Do you necessarily need it?

2) Implementation of fast nanoscale technique. Considering the KHz frequency of the cantilever, how can you measure nanosecond effects?

Moreover, the results you obtained, when compared to ODMR, are not so impressive (Contrast, SNR). This is understandable as it is a first and successful experiment of a new spin detection technique. But, can you estimate how much you could improve that? Do you expect results comparable to ODMR or PDMR? Can your technique be more effective than PDMR for some kind of defects in some materials? In that case, what are the specific characteristics of the defects and the surfaces that allow your technique to perform better than PDMR? Could you use this technique for ensembles?

Response 12.

We thank the reviewer for these important comments. We believe, that the presence of the cantilever is not necessary for observing the effects we report here, but this still has to be confirmed experimentally. Concerning the temporal resolution, currently the experiment is

indeed limited by the kHz frequency of the cantilever and also by the lock-in detection. For a faster measurement we would need to detect the surface voltage via a capacitive measurement, where nanosecond time scale can be achieved as demonstrated by T. Dittrich [11, 12]. In our experiments the contrast and SNR values are indeed lower compared to the ODMR or PDMMR, but these could be further improved and become comparable since they are not yet limited by the spin-to-charge conversion process [13].

At this stage we cannot really compare our method with PDMMR, since we have not performed both experiments on the same sample. The diamond surface was not specially treated, so we will still have to investigate how the surface termination affects the SVDMMR signal. We also demonstrate the detection of a non-fluorescent defect centre, and it would be interesting to see if it can be detected with a photocurrent-based techniques.

We see one of the advantages of the photovoltage-based detection over the photocurrent-based one in the fact, that PV measurements do not require a low contact resistance at the sample-electrode interface, so the quality of the surface electrodes is not as important.

Concerning the detection of ensembles, we believe that the surface charge might not scale linearly with the number of NVs since NV centers act as hole traps with a large capture cross-section (see for example [14]), thus blocking charge carriers from reaching the surface.

We have expanded the last paragraph of Discussion section of the manuscript to elaborate on the points and questions raised by the reviewer.

2.2 Comments to SI

Comment 1. SI2. Line 6. Why do you use that formula for fitting PV_{surf} ?

Response 1. The reason that we used this function to fit the PV_{surf} signal is the following. The measured PV signal is strong in the vicinity of the cantilever and weakens as the laser goes further away from the tip. To account for this change in the PV_{surf} , the surface PV signal in the vicinity of each of the NV centres was fitted a function representing a plane. We have added the following sentence to SI (S2 section) to make it clearer to readers:

The PV_{surf} was chosen to be a plane to account for a change in the PV signal with the distance to the cantilever.

Comment 2. Fig S2f. What do you mean by saturation in this plot? How can you say that the saturation is above 20 μ w (line 23, SI3)

Response 2. We believe, that the PV_{surf} signal is saturated in this graph, because even at the lowest laser power that we were able to measure, the PV_{surf} signal is already in the range of 0.8–1.05 V. At zero power (laser turned off) there is, of course, no photovoltage, so the rapid increase in the photovoltage from 0 V to ≈ 1 V takes place from 0 μ W to 20 μ W. Based on these results we assume, that additional holes are not able to reach the cantilever, as it is screened by the already trapped holes.

We have corrected the mentioned sentence in SI (S3 section): Since at a laser power above ~ 20 μ W the PV signal from the surface stays in the range of 0.8–1.05 V not showing a strong dependence on the laser power (see Fig. S2f), we assume, that at these laser powers the AC potential applied to the cantilever is screened by the already captured holes.

Comment 3. Fig S4c. Is the envelope and so the lock-in present also when MW are off?

Response 3. The envelope in our case determines whether the microwaves are reaching the sample. That is why in the case of Fig. S4c, the microwaves are present when the envelope

is on and not present when the envelope is off. In the case of Fig. S4d, the microwaves are already premodulated in Rabi pulses, which reach the sample when the envelope is on, and do not when the envelope is off. On the technical side it is done by an MW switch, which is controlled by a transistor-transistor logic (TTL) pulse representing the envelope.

We have added the following sentence to SI (S5 section) to make it clearer to readers:

This is done by an MW switch that transmits the MWs to the sample depending on a logic pulse (envelope).

Comment 4. Fig S4f: Do you find any dependence of the contrast and shape of the Rabi oscillations on laser power and pulse duration? How did you choose the laser power and pulse duration?

Response 4. We do not expect a strong dependence of the SVDMR and Rabi oscillations shape on the laser power. As mentioned in the main text and demonstrated in SI, the PV signal has a delay (or lag) compared to the optical PL. This delay may be reduced by increasing laser power. Therefore, higher laser power can be used to reduce the pulse block length (see Fig. S4c, e), decreasing the total accumulation time. However, high laser power can result in a decrease of the contrast, when the optical excitation rate becomes comparable to the Rabi frequency. Thus, the laser power was chosen to be high enough to reduce the length of the pulse block, but still have the maximum contrast, which we confirmed with conventional optically detected Rabi oscillations.

The laser pulse duration was chosen to be long enough to polarize the NV centre in the $|m_s = 0\rangle$ state from the one hand, and to be short enough to keep only the spin-dependent signal. Basing on our observations with the PL signal and the available literature (see, for example, [15]), it was chosen to be 1 μ s.

To provide this information to readers, we have indicated the laser power used for measurements (500 μ W for SVDMR and 750 μ W for the Rabi experiment) in the caption of Fig. 3 in the main text, and added the following sentences in SI (S5 section):

According to our observations, an increase in the laser power can result in a reduction of the signal settling time, mentioned above and clearly visible in Fig. S4d. Therefore, higher powers can be used to reduce the pulse block length (see Fig. S4c, e), decreasing the total accumulation time. However, high laser power can also result in a decrease of the contrast, when the optical excitation rate becomes comparable to the Rabi frequency. Thus, the laser power was chosen to be high enough to reduce the length of the pulse block, but still have the maximum contrast, which we confirmed with conventional optically detected Rabi oscillations. For typical measurements, the laser power of 500 μ W was used for SVDMR experiments, and 750 μ W for PV-detected Rabi oscillations. The laser pulse length for Rabi experiments was chosen to be 1 μ s, so that it is long enough to polarize the NV centre in the $m_s = 0$ state and short enough not to lose the spin contrast [15].

References

- [1] D. Bluvstein, Z. Zhang, and A. C. B. Jayich, “Identifying and Mitigating Charge Instabilities in Shallow Diamond Nitrogen-Vacancy Centers,” *Physical Review Letters*, vol. 122, p. 076101, Feb. 2019.
- [2] A. Karim, I. Lyskov, S. P. Russo, and A. Peruzzo, “Bright *ab initio* photoluminescence of NV+ in diamond,” *Journal of Applied Physics*, vol. 130, p. 234402, Dec. 2021.

- [3] A. Lozovoi, G. Vizkelethy, E. Bielejec, and C. A. Meriles, “Imaging dark charge emitters in diamond via carrier-to-photon conversion,” *Science Advances*, vol. 8, p. eabl9402, Jan. 2022.
- [4] F. M. Hrubesch, G. Braunbeck, M. Stutzmann, F. Reinhard, and M. S. Brandt, “Efficient Electrical Spin Readout of NV - Centers in Diamond,” *Physical Review Letters*, vol. 118, p. 037601, Jan. 2017.
- [5] A. Axt, I. M. Hermes, V. W. Bergmann, N. Tausendpfund, and S. A. L. Weber, “Know your full potential: Quantitative Kelvin probe force microscopy on nanoscale electrical devices,” *Beilstein Journal of Nanotechnology*, vol. 9, pp. 1809–1819, June 2018.
- [6] K. Bian, W. Zheng, X. Zeng, X. Chen, R. Stöhr, A. Denisenko, S. Yang, J. Wrachtrup, and Y. Jiang, “Nanoscale electric-field imaging based on a quantum sensor and its charge-state control under ambient condition,” *Nature Communications*, vol. 12, p. 2457, Apr. 2021.
- [7] W. Zheng, K. Bian, X. Chen, Y. Shen, S. Zhang, R. Stöhr, A. Denisenko, J. Wrachtrup, S. Yang, and Y. Jiang, “Coherence enhancement of solid-state qubits by local manipulation of the electron spin bath,” *Nature Physics*, vol. 18, pp. 1317–1323, Nov. 2022.
- [8] A. Wood, A. Lozovoi, Z.-H. Zhang, S. Sharma, G. I. López-Morales, H. Jayakumar, N. P. De Leon, and C. A. Meriles, “Room-Temperature Photochromism of Silicon Vacancy Centers in CVD Diamond,” *Nano Letters*, vol. 23, pp. 1017–1022, Feb. 2023.
- [9] H. Jayakumar, A. Lozovoi, D. Daw, and C. Meriles, “Long-Term Spin State Storage Using Ancilla Charge Memories,” *Physical Review Letters*, vol. 125, p. 236601, Dec. 2020.
- [10] E. Bourgeois, M. Gulka, and M. Nesladek, “Photoelectric Detection and Quantum Readout of Nitrogen-Vacancy Center Spin States in Diamond,” *Advanced Optical Materials*, vol. 8, p. 1902132, June 2020.
- [11] T. Dittrich, S. Fengler, and M. Franke, “Transient surface photovoltage measurement over 12 orders of magnitude in time,” *Review of Scientific Instruments*, vol. 88, p. 053904, May 2017.
- [12] T. Dittrich, “Transient surface photovoltage spectroscopy of diamond,” *AIP Advances*, vol. 12, p. 065206, June 2022. Number: 6.
- [13] B. Shields, Q. Unterreithmeier, N. De Leon, H. Park, and M. Lukin, “Efficient Readout of a Single Spin State in Diamond via Spin-to-Charge Conversion,” *Physical Review Letters*, vol. 114, p. 136402, Mar. 2015.
- [14] A. Lozovoi, H. Jayakumar, D. Daw, G. Vizkelethy, E. Bielejec, M. W. Doherty, J. Flick, and C. A. Meriles, “Optical activation and detection of charge transport between individual colour centres in diamond,” *Nature Electronics*, vol. 4, pp. 717–724, Oct. 2021. Number: 10.

- [15] M. Gulka, D. Wirtitsch, V. Ivády, J. Vodnik, J. Hruby, G. Magchiels, E. Bourgeois, A. Gali, M. Trupke, and M. Nesladek, “Room-temperature control and electrical readout of individual nitrogen-vacancy nuclear spins,” *Nature Communications*, vol. 12, p. 4421, July 2021.

Voltage detected single spin dynamics in diamond at ambient conditions

We would like to thank again both reviewers for evaluating our manuscript. We have taken seriously their advice and suggestions, and we have thoroughly revised the text. Below you can find a detailed reply to the critics raised by the reviewers, followed by a list of the changes we have made in the text. We have also included a brief summary of the main findings of the manuscript.

1 Response to Reviewer #1

Comment 1. One minor point to make is towards the author’s citation of Ref 32 (Peruzzo paper 2021) regarding NV+ PL: This paper is a bit of an outlier, I think it’s now pretty well established NV+ is optically dark. At least no one has experimentally reported NV+ PL that I am aware of.

Response 1. We agree with Reviewer 1. We also could not find an experimental paper reporting photoluminescence (PL) from NV⁺.

We, therefore, followed the Reviewer’s suggestion and corrected the sentence “As shown by *ab-initio* calculations in [1], the luminescence spectrum of a positively charged NV centre is below 1.65 eV in photon energy, or above 775 nm in photon wavelength, therefore out of our detection range 650–750 nm, which could explain the absence of the luminescence signal.” to :

“To our knowledge, there is no experimental demonstration of PL of an NV center in the NV⁺ charge state, which could explain the absence of the luminescence signal.”

2 Response to Reviewer #2

Comment 1. In the main text, you define the imaging contrast (eq1) to evaluate the visibility of the NV center with respect to the background, but you do not use this parameter elsewhere in the text. Then you define in eq2 the magnetic resonance contrast that you use for quantitative measurements all along the text. I will suggest, but the choice is up to you, not to call “imaging contrast” the parameter of eq1 and so to use a different letter for this parameter. I think that can make easier the reading of the paper.

Response 1. We agree with the reviewer that we don't use this imaging contrast in the main text. As we define it also in the supplementary Information (SI), we have decided to remove it from the main text. And moved the following sentence from the main text to SI (after Supplementary Eq. 1):

“In this equation we use absolute values in the denominator, since the PV_{NV} and PV_{surf} signals have different signs.”

Comment 2. Page 8: “The NV PV signal also shows a dependence on the laser power (see Fig.:S2e), where the saturation of is presumably caused by the screening of the applied VAC by the built-up charge and/or low density of surface states, compared to the local carrier concentration.” Without reading the supplementary material, I think the reader cannot understand what saturation you are talking about. Maybe you can rephrase a little bit this sentence introducing the fact that the NV PV signal saturates increasing the laser power.

Response 2. We thank the reviewer for this comment. We have rephrased this sentence to:

“The NV PV signal also shows a dependence on the laser power (see Supplementary Fig. 3a), where the saturation of the signal at higher powers is presumably caused by the screening of the applied V_{AC} by the built-up charge and/or low density of surface states, compared to the local carrier concentration.”

Comment 3. In the SI, S2, it is not clear if the fitting model for PV surface is based on an expected linear behaviour or, after having tested different models, the linear is the one that best fit your results. It could be interesting to add an image showing the fit, as you do for PV NV in figS2c.

Response 3. We have approximated the background PV with a linear function because on the one hand, the PV map shows a change in PV_{surf} throughout the image, so the constant background is not suitable. On the other hand, more complex background function will interfere with the PV_{NV} signal. Therefore, we have decided to use a linear function as it is the simplest function that gives the needed effect of the background gradient.

We have added the PV_{surf} map to Supplementary Fig. 2d.

Comment 4. Figure S4a-b, you compare the CW and the pulsed acquisition procedure. It would be better, in my opinion, to use the same parameter on the Y-axis to make the comparison.

Response 4. We agree with the reviewer, that it would be better to compare the same parameter. However, for this one would need to record the PV signal without application of microwaves during the measurements, which we did not do. However, for a better comparison of the amplitude of the signal, one can use PV signal at a nonresonant frequency as a baseline instead. We have decided to use this approach and changed the mentioned graph and added the following text to its description.

“To be able to compare signal amplitudes in both modes, in Supplementary Fig. 5a we have plotted $\Delta\widetilde{PV}^{MR} = PV(\nu) - PV(\nu_{off.res})$ and $\Delta\widetilde{PL}^{MR} = PL(\nu) - PL(\nu_{off.res})$, where $PV(\nu_{off.res})$ ($PL(\nu_{off.res})$) is the PV (PL) signal at a non-resonant (off-resonant) frequency. In our case, we have used an average of the PV and PL signals at the first two frequency points (2.85 and 2.852 GHz) for estimation of these values. ”

References

- [1] A. Karim, I. Lyskov, S. P. Russo, and A. Peruzzo, “Bright *ab initio* photoluminescence of NV+ in diamond,” *Journal of Applied Physics*, vol. 130, p. 234402, Dec. 2021.